# From Pre- and Probiotics to Post-Biotics: A Narrative Review [note 1]

**DOI:** 10.3390/ijerph19010037

**Published:** 2021-12-21

**Authors:** Emidio Scarpellini, Emanuele Rinninella, Martina Basilico, Esther Colomier, Carlo Rasetti, Tiziana Larussa, Pierangelo Santori, Ludovico Abenavoli

**Affiliations:** 1Internal Medicine Unit, “Madonna del Soccorso” General Hospital, 63074 San Benedetto del Tronto, Italy; martina.basilico@sanita.marche.it (M.B.); carlo.rasetti@sanita.marche.it (C.R.); pierangelo.santori@sanita.marche.it (P.S.); 2TARGID, KU Leuven, 3000 Leuven, Belgium; esther.colomier@kuleuven.be; 3UOC di Nutrizione Clinica, Dipartimento di Scienze Mediche e Chirurgiche, Fondazione Policlinico Universitario A. Gemelli IRCCS, Largo A. Gemelli 8, 00168 Rome, Italy; emanuele.rinninella@unicatt.it; 4Dipartimento di Medicina e Chirurgia Traslazionale, Università Cattolica Del Sacro Cuore, 00168 Rome, Italy; 5Department of Health Sciences, Magna Græcia University, 88100 Catanzaro, Italy; l.abenavoli@unicz.it

**Keywords:** gut microbiota, prebiotics, probiotics, symbiotic, postbiotics

## Abstract

Background and aims: gut microbiota (GM) is a complex ecosystem containing bacteria, viruses, fungi, and yeasts. It has several functions in the human body ranging from immunomodulation to metabolic. GM derangement is called dysbiosis and is involved in several host diseases. Pre-, probiotics, and symbiotics (PRE-PRO-SYMB) have been extensively developed and studied for GM re-modulation. Herein, we review the literature data regarding the new concept of postbiotics, starting from PRE-PRO-SYMB. Methods: we conducted a search on the main medical databases for original articles, reviews, meta-analyses, randomized clinical trials, and case series using the following keywords and acronyms and their associations: gut microbiota, prebiotics, probiotics, symbiotic, and postbiotics. Results: postbiotics account for PRO components and metabolic products able to beneficially affect host health and GM. The deeper the knowledge about them, the greater their possible uses: the prevention and treatment of atopic, respiratory tract, and inflammatory bowel diseases. Conclusions: better knowledge about postbiotics can be useful for the prevention and treatment of several human body diseases, alone or as an add-on to PRE-PRO-SYMB.

## 1. Introduction

The human microbiota, now considered as a “functional organism“, consists of a complex community of microorganisms (bacteria, yeasts, fungi, archea, protozoa, and viruses), living on our skin and mucosal tissues, hence forming an efficient ecosystem together with the body [1,2].

In humans, gut microbiota is crucial in metabolism, immune tolerance, and nutrients absorption, within and outside the gut [3]. Thus, an imbalance of gut microbiota, generated by diet, use of antibiotics, or infections, can be detrimental for host equilibrium [4].

The latter, namely “dysbiosis”, can express as intestinal and extra-intestinal diseases [5]. The first group includes liver steatosis (NAFLD) until the liver cirrhosis stage [5], inflammatory bowel diseases (IBD), irritable bowel syndrome (IBS), celiac disease, and gastro-intestinal cancers [4]. The second group includes obesity, atopic and autoimmune diseases, autism, and systemic sclerosis [4].

The main measures to re-establish unbalanced gut microbiota are the use of diet, antibiotics, pre-, and probiotics [6]. However, emerging evidence shows how the use of living organisms, namely probiotics, is not devoid of virulence emergence and antibiotic-resistance development over time [6]. Thus, there is a strong need for use of safer and equally effective gut microbiota modulatory agents. One very promising chance is represented by bacterial products and the components of probiotics, namely “postbiotics” [7].

This review summarizes the literature data on nomenclature, the description of emerging postbiotics. We further describe their potential mechanisms of action and use in disease prevention/treatment. Finally, due to the tough times we are living in because of the SARS-CoV-2 pandemic, we briefly describe the potential use of postbiotics in COVID-19 patients.

## 2. Materials and Methods

We conducted a PubMed and Medline search for original articles, reviews, meta-analyses, and case series using the following keywords, their acronyms, and their associations: gut microbiota, prebiotics, probiotics, symbiotic, and postbiotics. When appropriate, preliminary evidences from abstracts belonging to the main national and international gastroenterological meetings (e.g., United European Gastroenterology Week, Digestive Disease Week) were also included. The papers found from the above- mentioned sources were reviewed by two of the authors (L.A. and E.S.), according to PRISMA guidelines [8].

The last MEDLINE search was dated 30 October 2021.

## 3. Results

### 3.1. Gut Microbiota Composition and Main Functions

The human GI tract hosts over 100 trillion microbes, the vast majority being bacteria [3].

The latter are divided in phyla, classes, orders, families, genera, and species [9]. Main gut microbial phyla are: Firmicutes, Bacteroidetes, Actinobacteria, Proteobacteria, Fusobacteria, and Verrucomicrobia. However, the two phyla Firmicutes and Bacteroidetes account for almost 90% of the entire gut microbiota. Firmicutes includes more than 200 different genera (e.g., Lactobacillus, Bacillus, Clostridium, Enterococcus, Ruminicoccus, and Clostridium) and Bacteroidetes has two of the most abundant genera (Bacteroides and Prevotella) [1,9].

The entire genome of the gut microbiota is called “microbiome” and is about 150-fold bigger than that of a human cell [10].

One-tenth of the total colonizing bacterial species per individual constitutes a dynamic “microbial fingerprint”: ageing, dietary changes, and use of antibiotics, prebiotics, and probiotics can make it fluctuate [11]. Indeed, there is an intestinal microbial ‘core’ that includes 66 species conserved in over 50% of the general population [12].

The use of culture-based methods has limited the study of gut microbiome composition. However, the advent of new metagenomic technologies has paved the way to the definition of inter/intra-individual variability of the gut microbiome [11].

Upon delivery, microbial life starts with a limited and unstable repertoire of microorganisms that evolves and adapts to environmental changes during our lifespan [13,14,15].

Gut microbiota is crucial for nutrient absorption and fermentation, regulation of intestinal permeability (IP), host metabolism (e.g., carbohydrates absorption and processing, proteins putrefaction, bile acids formation, insulin sensitivity), and modulation of intestinal and systemic immunity. This is crucial in childhood and adulthood, for the maintenance of antigen tolerance and the containment of pathogens expansion [16].

Thus, there is ever-growing interest in the remedies and approach to re-establish gut microbiota composition through diet and antibiotics as well as with pre- and probiotics [3,17].

### 3.2. Pre-, Probiotics and Symbiotics

Prebiotics are food components beneficially affecting gut microbiota. The main examples of their group are human milk oligosaccharides, lactulose, fructo-oligosaccharides, and inulin [18]. Their impact on gut microbiota is relevant. They have been extensively used in functional food manufacturing and as an add-on treatment for dysbiosis [18]. However, adjusted dose-finding and treatment-duration studies are needed to evaluate and assess the optimal scheme of administration [18].

Prebiotics are very often administered together with probiotics, namely living microorganisms (e.g., bacteria, yeasts), beneficially affecting host health [17]. Several studies on probiotics have been analyzing their real impact on human health and diseases [4]. More robust data vs. prebiotics are available on their efficacy and dose- and strain-effect relationship [19]. Pre- and probiotics, namely symbiotics, can be used together with an empowering mutual efficacy and multi-effect profile [19].

The spectrum of human diseases where pre- and probiotics are usable alone or in combination encompasses GI (e.g., NAFLD, cancer, celiac disease, IBD, IBS, diarrhea, functional dyspepsia, constipation) and extra-intestinal diseases (obesity, diabetes, autoimmune disease, atopic conditions, neurological disorders, psychiatric disorders) [19].

However, use of probiotics is not devoid of potential harmful issues, such as antibiotic resistance development as well as the growth and selection of virulent strains for humans [19].

### 3.3. A New Concept among the “Bugs”: Postbiotics

Postbiotics are defined as beneficial substances, resulting from microbiota metabolism and having a beneficial effect on the microbiota itself as well as the host [7,20]. Therefore, after looking extensively into the real origin of “the egg or the chicken”, and landing on the probiotics concept as the long-lasting “chicken”, we arrived at the discovery and usage of “the egg”.

There is still debate about whether this exclusive definition is accurate enough. For example, Tsilingiri et al. consider postbiotics “any substance released by or produced through the metabolic activity of the microorganism, which exerts a beneficial (direct or indirect) effect on the host” [21]. Salminen et al. claimed the concept of postbiotics to be inclusive of inactivated microorganisms beneficially affecting host health [20]. However, this definition has not been accepted by the rest of the scientific community [22]. Thus, we assume that postbiotics include all substances of bacterial/fungal origin with a beneficial effect on the host. In addition, they fail to meet the criteria for probiotics and prebiotics [22].

Main representatives of postbiotics are: cell-free supernatants, exopolysaccharides, enzymes, cell wall fragments, short chain fatty acids (SCFA), bacterial lysates, and gut microbiota metabolites.

#### 3.3.1. Supernatants

Cell-free supernatants are composed by active metabolites produced by bacteria and yeast into the surrounding liquid. They can be manifactured directly from cell cultures. In detail, after an incubation period, the microbes are centrifuged and then removed. Therefore, the supernatant mixture is filtered for sterility issues.

Different bacterial strains have different supernatants with different functions. For example, *Lactobacilli acidophilus* and *L. casei* supernatants have anti-inflammatory and antioxidant effects, respectively [9]. Moreover, supernatants derived from *L. casei* and *L. rhamnosus* GG cultures have been shown to prevent the invasion of colon cancer cells [10] and to reduce the oxidative stress in vivo [23].

Supernatants derived the genera Lactobacillus and Bifidobacterium were also recently shown to have antibacterial activity. For example, they were able to prevent the invasion of enteroinvasive *E. coli* strains into enterocytes in vitro [24]. Indeed, this antibacterial capability may result from the inhibition of adhesion of the pathogens, strengthening of the cell barrier, and improved expression of the protective genes [24] (Table 1). *L. plantarum* supernatants have a trophic effect on the structure of the intestinal barrier [25]. In fact, administration of these supernatants to lambs early in life leads to an improved absorption surface of the intestine and a decrease in intestinal pathogens [25].

Finally, supernatants from two well-known yeasts, namely *Saccharomyces cerevisiae* and *Saccharomyces boulardii*, were shown to be able to reverse the impaired intestinal peristalsis stress-induced [26]. Interestingly, *S. boulardii* supernatants have anti-inflammatory and antioxidant activity, with improved wound healing and also promising implications for rebuilding the intestinal barrier [7,27].

#### 3.3.2. Exopolysaccharides

Microorganisms produce bio-polymers with several chemical properties. They can be released outside the bacterial cell wall, forming the variegate complex of exopolysaccharides (EPSs). EPSs are not a new postbiotic as they are already used in the food industry as stabilizing, emulsifying, and water-binding agents [28]. EPSs may also have newer immunomodulatory properties through the interaction with dendritic cells (DCs) and macrophages, resulting in enhanced proliferation of T and NK lymphocytes [29]. For example, EPS derived from *L. casei* can be used as adjuvant for improvement of the efficacy of the foot-and-mouth disease vaccine [30].

The antioxidant potential of an EPS obtained from *L. helveticus*, namely uronic acid, contained also in green tea, is linked to its iron-binding properties [31]. Among the variegate properties of EPSs there is the inhibition of cholesterol absorption [32]. Indeed, consumption of kefiran, EPS produced by *L. kefiranofaciens*, has shown efficacy delaying the development of atherosclerosis in an animal model [33].

Another class of EPS, β-glucans may enhance the cell-mediated immune response against bacteria, viruses, parasites, and cancer cells [34,35]. Interestingly, β-glucans may also potentiate probiotics effects through a facilitated adhesion of lactobacilli to the intestinal wall [36]. Similarly, they can increase the bioavailability and absorption of carotenoids in the GI-tract [37]. Finally, topic application of β-glucans can help improve healing from atopic dermatitis and prevent it from relapsing [38] (Table 1).

#### 3.3.3. Antioxidant Enzymes and “Bile Salts Hydrolase Case“

Born as defensive mechanism of cells, antioxidant enzymes (namely, glutathione peroxidase (GPx), peroxide dismutase (SOD), catalase, and NADH-oxidase) are crucial for microorganism survival. Two strains of *L. fermentum* were found to have a high content of GPx [39], with strong antioxidant action in vitro [40]. *L. plantarum* was also demonstrated to have increased GPx concentration [41]. Interestingly, genetically modified Lactobacilli strains synthetizing SOD/catalase showed capability to relieve symptoms of a mouse model of Crohn’s disease vs. “wild-type” strains [42] (Table 1).

Conjugated bile salts selectively regulate GM composition, in general, and Lactobacilli-one, in particular, within the gut [43]. For example, some Lactobacilli may confer protection against the well-known Giardia lamblia parasite, from in vitro and in vivo evidences. In detail, *L. johnsonii* and, more recently, *L. gasseri* CNCM I-4884 strain bile salts hydrolases are able to generate deconjugated bile salts toxic to the parasite [44]. Thus, these hydrolases can have a prophylactic use against this parasite.

#### 3.3.4. Cell Wall Components

Bacterial cell wall parts are able to trigger a specific immune response. One of the well-known components is bacterial lipoteichoic acid (LTA) of gram-positive bacteria [45]. Its immune-modulatory effect is complex: some reports indicate LTA reducing IL-12 production and inducing the production of immunomodulatory cytokines [46]; some others show LTA not alleviating inflammation, with damage to intestinal tissues [47] (Table 1).

However, LTA use in dermatology is clearer: topical LTA application enhances and leads to an anti-infectious peptides release (e.g., β-defensin and cathelicidin) [48]. In addition, Lactobacilli and Bifidobacteria produce significant amounts of LTA that stimulate skin mast cell response vs. bacterial and viral infections [49]. More interestingly, LTA has also anti-cancer activity [50,51]. However, further safety evaluation for LTA use is warranted.

#### 3.3.5. SCFAs

SCFAs are well-known products of fermentation of plant polysaccharides by gut microbiota. They include acetic, propionic, and butyric acids as well as the corresponding fatty acid salts.

Butyrate is the most important energy source and trophic agent for enterocytes. However, butyrate has also shown immunosuppressive properties [52] as is able to induce food tolerance. This effect is due to its capability to increase the expression of some immunosuppressive cytokines such as type 1 interferons [IFNs], IL-10, and TGF-β [52]. In fact, rectal administration of butyrate results in significant regression of inflammation of the colon of patients with ulcerative colitis vs. a placebo [53].

Interestingly, intestinal colonization by Roseburia intestinalis, one of the major producers of butyrate in the intestine, is able to block atherogenesis in a mouse model [54].

SCFAs can contribute to energy harvesting through stimulation of G-protein coupled receptors (GPCRs) and secretion of the glucagon-like peptide 1 (GLP-1). Indeed, increased serum and fecal levels of acetate is associated with improved insulin sensitivity and related reduction of fat deposition in vivo [55]. More interestingly, acetate can also direct food intake through appetite down-regulation in the central nervous system [56].

A diet rich in acetate significantly increased resistance to enterohaemorrhagic *E. coli* O157:H7 infection in a mouse model [57].

Propionate is one of the main substrates for gluconeogenesis within the liver. In addition, it has a “statin-like” effect due to the inhibition of cholesterol synthesis [58]. Propionate also shows an in vivo anti-inflammatory activity comparable to that of butyrate [59].

Indeed, the ratio of SCFAs used is important for cholesterol homeostasis maintenance.

A therapeutic or preventive perspective of SCFAs use in neurology derives from the evidence of symptom relief using an animal model of multiple sclerosis [60].

#### 3.3.6. Bacterial Lysates

Bacterial lysates (BLs) are obtained by the chemical/mechanical degradation of gram-positive and -negative bacteria. Their safe clinical use is a consolidated practice in pediatrics [61] as well as in internal and infectious disease medicine. The rationale for their use in viral/bacterial infection prevention relates to the concept of “gut-lung axis” that describes the interplay between the enteric system (namely, GALT) and the respiratory immune system [62]. Several mechanistic studies show orally administered lyophilized BLs reaching Peyer’s patches in the small bowel, with stimulation of dendritic cells (DCs) and activation of T and B lymphocytes [61] that migrate within the mucous membrane of the respiratory tract. This results in innate immune system stimulation and IgA secretion [63].

Thus, the extreme “hygiene” of the western and “westernized” world can be bypassed by using BLs. The case of BLs is of special interest for clinicians, as their use allows for the mimicry of the presence of bacteria in order to stimulate the immune system, without the harmful effects of their actual presence. Therefore, we can assume that they work as “mild and broad vaccine“.

Robust data support these observations: a 2018 meta-analysis including more than 4800 children highlighted a significantly lower incidence of respiratory tract infections in those receiving BLs vs. a control group [64]; a 2020 systematic review showed BLs as an add-on treatment to be effective in the reduction of wheezing episodes and the frequency of asthma exacerbations in children [65,66,67].

#### 3.3.7. Metabolites

This section describes a variegate of molecules produced by bacteria. They include vitamins, phenolic-derived metabolites, and aromatic amino acids. They are highly bioavailable, possessing antioxidative and signaling effects. Thus, they are crucial in the bug-host “cross-talk” (Table 1).

Intestinal-bacterial-produced folate may have both beneficial and detrimental effects. On one hand, folate supplementation leads to a lower risk of stroke vs. controls [68]. On the other hand, folate supplementation or hyper-production can accelerate carcinogenesis in subjects at risk of, and patients with, colorectal cancer. Thus, from a clinical point of view this postbiotic use can show a “U-shaped relationship” with health and disease [69].

Vitamin B12 and the other B complex vitamins de novo synthetized by intestinal bacteria are a potential source for the host [70]. In fact, an *L. acidophilus* add-on to a yogurt matrix has been associated with increased synthesis of vitamin B12 and, more importantly, reduced prevalence of anemia [71].

In humans, vitamin K maintains normal blood coagulation and there are different iso-forms with different anti-coagulant activity. Some forms of vitamin K2, namely menaquinones, are present in ingested food such as cheese, natto (a Japanese food), and curd. Interestingly, they are also produced by some bacteria genera of GM, as most intestinal anaerobic and aerobic gram-positive bacteria use menaquinones in their electron transport pathways. Some menaquinone-producing bacteria are: Eubacterium lentum, which produces MK-6; *Lactococcus lactis ssp. lactis* and *spp. cremoris* that mainly produce MK-8 and MK-9; and Bacteroides fragilis, which produces MK-10, MK-11, and MK-12 [72]. Finally, K2-vitamin form, produced also by *L. casei*, has shown some promising anti-cancerogenic properties [73].

GM is involved in the synthesis and metabolism of aromatic amino acids (AAA) that act as “biochemical bullets” on the kidneys, brain, and cardiovascular system [74]. In fact, genetic manipulation of GM-synthetizing indoxyl sulfate contributes to the progression of chronic kidney disease [75].

Dietary polyphenols are actively metabolized by gut microbiota. Thus, evidence from different metabolic phenotypes supported the hypothesis of the existence of different “metabotype” deriving from dietary components and GM interactions [76]. This is one of the mainstays of the “personalized nutrition” build-up [77]. In detail, postbiotics derived from dietary polyphenols include urolithin A (UA), equol, and 8-prenylnaringenin (8-PN). Mice administered with UA for 10 weeks lost 23.5% more weight vs. controls with consensual improvement of insulin resistance [78,79].

Intriguingly, middle-aged Japanese women supplemented with equol for one year showed a significant carotid arterial stiffness reduction, an improvement of the cholesterol profile [80]. Even more interestingly, the same scheme of administration led to a whole body bone mineral density increase in postmenopausal women [81].

Bioactive peptides are oligopeptides consisting of 2–20 amino acids. They derive from food proteins such as milk, egg, fish, rice, soybean, pea, chlorella, spirulina, oyster, and mussel. They have mainly immunomodulatory and anticancer activities [82]. These peptides can be produced using various methods: in vitro enzymatic hydrolysis, autolytic process using endogenous enzymes, and, last but not least, microbial fermentation. The latter is operated mainly by Lactobacilli with their proteases. Among the most interesting and clinically relevant biopeptides, we can find milk-derived peptides. They have complex immune-modulating activities via cytokines regulation, resulting in attenuation of allergic reactions both in animal and preliminary in vitro and in vivo studies [83,84].

### 3.4. Actual and Future Applications of Postbiotics

Although postbiotics can be considered pleiotropic agents on humans’ health, there are different effects that deserve a deeper description.

#### 3.4.1. Immunomodulation and Anti-Cancer Effects

Immunomodulatory effects of postbiotics have been reported in the last 30 years. In detail, SCFAs, and, more in detail, propionate, are able to up-regulate Tregs [85]. Supernatant, cell wall fragments from Bacillus coagulans culture also promote T helper (Th)2-dependent immune response [86]. In addition, supernatant from Bifidobacterium breve culture is able to limit the Th1-mediated and enhance Th2-mediated responses [87,88]. These effects are often observed in mice models of atopic diseases [88].

The SCFA propionate is able to selectively induce apoptosis in gastric cancer cells [89]. Interestingly, SCFAs can also modulate onco- and suppressor genes expression through epigenetic modifications: *L. rhamnosus* GG supernatant increases ZO-1 expression (responsible for intercellular permeability) and decreases MMP-9 expression that facilitates cancer cell penetration [7] (Table 1).

#### 3.4.2. Anti-Infectious Effects

Some postbiotics can competitively bind to receptors for pathogenic bacteria, are able to change the expression of host genes, and modulate host environmental components [90]. Starting from this paradigm, it is interesting to note that the newer combination of postbiotics and probiotics can efficaciously prevent rotavirus-associated diarrhea [91]. However, this finding about a new kind of “symbiotic” is only preclinical. In a group of children aged 12–48 months, the daily intake of *L. paracasei* postbiotic leads to reduction in the incidence of diarrhea [92], acute gastroenteritis, pharyngitis, laryngitis, and tracheitis [93].

#### 3.4.3. Metabolism Modulation and Anti-Atherosclerotic Effects

Postbiotics can regulate and reshuffle lipid metabolism. This can result in a significant reduction of cardiovascular risk and related accidents. Propionate is well-known, having a “statin-like” effect (similarly to nutraceuticals, e.g., curcumin, K-monacolin) [94]. It is interesting to note that the case of kefiran conjugates antiatherogenic (e.g., prevention of cholesterol accumulation in macrophages and reduction of lipid concentration) and anti-inflammatory actions [95]. *Lactobacilli BLs* reduce the levels of triglycerides and LDL cholesterol, while increasing the level of beneficial HDL cholesterol in an obese mouse model [96]. These effects are explained by activation of the peroxisome proliferator-activated receptor (PPAR), the same therapeutic target of fibrates [96].

#### 3.4.4. Detoxification and Wound Healing Effects

Autophagy is a self-degradative process, cleaning out cells and their components from tissues. It is an efficacious response to various stress stimuli such as those from diet. Bacterial peptidoglycan promotes autophagy through the NOD1 receptor [97]. More in detail, *L. fermentum* postbiotics trigger autophagy in hepatic cells HepG2, resulting in a protective effect towards induced liver toxicity [98]. Furthermore, urolithin A inhibits mitophagy, namely autophagy of mitochondria, and can potentially prevent or delay muscle ageing [99].

Non-experts often ignore that oxytocin, an important gynaecologic neuropeptide, can also stimulate and accelerate wound healing. BLs obtained by sonication of *L. reuteri* significantly increase the number of oxytocin-producing cells in the hypothalamic periventricular nuclei, raising hormone blood concentration in animal models [100]. The subsequent probiotic-generating postbiotic administration to both animal and human models confirmed these results, with a good safety profile [100].

#### 3.4.5. Functional Foods Preparation

Functional foods (FF) resemble dietary components with a clear nutritional value and other beneficial health effects. Postbiotics and probiotics are already part of the preparation of functional foods.

One of the main advantages of FF enrichment by postbiotics is the host’s immune-stimulation. For example, the cell-free fraction of fermented milk is able to prevent Salmonella infection in mice [101]. Furthermore, postbiotics from B. breve and Streptococcus thermophilus are currently used in the production of modified milk, in order to obtain a long-lasting reduction in the incidence of food intolerance and/or respiratory allergy in the first months of the life of children [102]. Importantly, only mild diarrhea was recorded in these trials [103].

#### 3.4.6. Future Perspectives and the COVID-19 Issue

As postbiotics are able to beneficially affect the maturation of the immune system, improve the regulation of intestinal permeability, and, last but not least, modulate gut microbiota composition, their future uses are expanding. They could be used to prevent and, further, treat several diseases devoid of an efficacious cure, such as Alzheimer’s disease and multiple sclerosis. Preliminary reports on the first clinical trials on the use of postbiotics for the abovementioned diseases seem promising [7].

Gut microbiota modulation with probiotics seem to affect the inflammatory storm of COVID-19 and to directly counteract replication of SARS-CoV-2 [104]. Thus, GM re-modulation via postbiotics can be crucial in preventing SARS-CoV-2 infection in predisposed humans [105]. In addition, GM remodulation can change the natural story of COVID-19 towards a milder form [105]. Similarly to probiotics, postbiotics could directly inhibit SASR-COV-2 replication.

In an in vitro and in silico study by Rather et al., an extract from *L. plantarum* Probio-88 (P88-CFS) was able to significantly inhibit the replication of SARS-CoV-2. In addition, P88-CFS-treated cells showed a significant reduction of inflammatory cytokines. Based on the in silico molecular docking data, it was unraveled that the antiviral activity of this strain derives from plantaricin E (PlnE) and F (PlnF), which is able to bind on the SARS-CoV-2 helicase. Thus, these postbiotics can act as a “blocker” of viral ss-RNA during its replication. This strain and/or its postbiotic substances can be used as an integrative approach, along with a vaccine, to contain the spread of SARS-CoV-2 and, most importantly, its variants [106].

A special mention is deserved for the emergence of “biological doping”. GM modulation can change humans’ metabolism and the availability of energy sources for a better sport performance [7]. For example, an increased abundance of Veillonella in our gut, able to metabolize lactic acid to propionate, significantly increases animals’ physical performance [106]. Similarly, enteral administration of propionic acid leads to similar results [106]. Another issue is represented by the recognition of this form of doping, which is emerging as an efficient road to skipping health controls for athletes.

## 4. Conclusions

After almost 40 years of “puzzle-making” about our gut microbiota composition, products of bacteria and their metabolites have found a nomenclature as “postbiotics”. Knowledge on their usability for the improvement of human health has rapidly grown in the last ten years. They can prevent/treat atopic, immune-mediated respiratory, and gastrointestinal diseases. They can change human metabolism until “biological doping” acquires harmful usability.

From an economic point of view, postbiotics have several pros: long life, easy storage, and a reduced need to maintain at a low temperature vs. pro- and symbiotics.

From a safety point of view, postbiotics lack the issue of antibiotic-resistance gene development, such as acquiring virulence factors for probiotics [101]. The latter have the emerging issue of living microorganism exposure to an immature immune system and a relatively “leaky” intestinal barrier, especially in children [7].

Finally, the potential usage of postbiotics in the frame of GM re-modulation paves the way for them to be set up as vaccines and real “silver bullets” for COVID-19 and other infectious diseases.

Strong industry affords are required to fulfill an accurate case-control development of the therapeutic field of postbiotics for action in human health.

## Figures and Tables

**Table 1 ijerph-19-00037-t001:** Main representatives of postbiotics and their functions.

Type of Postbiotic	Experimental Model and Number of Subjects in Study	Mechanism of Action	Effects	Overall References
**Exopolysaccharides**	Mice, *n* = 88 [29]	Interaction with dendritic cells and macrophages;	Immunomodulatory, vaccine adjuvants; Delay of atherosclerosis development	[28,29,30,31,32,33,34,35,36,37,38]
	In vitro [31]			
	rabbits, *n* = 7/8 (control and treatment group, respectively) [33]			
	in vitro [34]	inhibition of cholesterol absorption;		
	mice and in vitro [35];			
	in vitro [36]			
	rats, *n* = 6/8 and humans, *n* = 16	enhancement of immune response vs. pathogens		
	RCT [37];			
	105 humans [38]			
**Enzymes**	In vitro [39];	Antioxidant action	Potential relief of Crohn symptoms	[39,40,41,42,43,44]
	In vitro [40];			
	Post-weaning male lambs, *n* = 12 and in vitro [41];	gut microbiota modulation;		
	mice [42];			
	in vitro [43];	protection against pathogens (e.g., Giardia lamblia)		
	in vitro [44]			
**SCFA**	In vitro and 6 + 8 colitis model mice [52];	Energy source; immunosuppressive properties; energy harvesting and reduction of fat deposition; Inhibition of cholesterol synthesis	Ulcerative colitis regression; Atherosclerosis blocking; Improved insulin sensitivity; “statin-like effect“	[52,53,54,55,56,57,58,59,60]
	ulcerative colitis patients, *n* = 11 and ex vivo [53];			
	147 + 195 (male/female) mice [54];			
	mice, *n* = 12–12−12 and 11–12, for experiment 1 and 2, respectively [55];			
	mice, *n* = 12 + 12 and ex vivo [56];			
	mice [57];			
	in vitro [59];			
**Bacterial lysates**	232 children, multicenter RCT [61]; mice and in vitro [62]; 152 children, RCT [66]	Stimulation of dendritic cells and activation of T and B lymphocytes	Viral/bacterial disease prevention in children and adults; Reduction of asthma and wheezing episodes	[61,62,63,64,65,66,67]
**Supernatants**	In vitro and in vivo [23]; in vitro [24]; Twelve newly weaned lambs, randomized [25]; Swiss Webster mice and ex vivo [26]; in vitro [27]	Anti-inflammatory and antioxidant actions; prevention of invasion of colon cancer cells; antibacterial activity; trophic action on the intestinal barrier; reverse of the impaired intestinal peristalsis induced by stress	Prevention of enteroinvasive *E. coli* strains invasion into enterocytes in vitro; improvement of absorptive surface of the intestine and reduction of intestinal pathogens in lambs; wound healing	[23,24,25,26,27]
**Metabolites (vitamins, phenols, aromatic aminoacids, bioactive peptides)**	children [71]	Increased production of folate, B12, K vitamin, biopeptides.	Lower risk of stroke	[68,69,70,71,72,73,74,75,76,77,78,79,80,81,82,83,84]
	mice, *n* = 6 + 6 + 6 + 6, randomized [73]		Reduction of anemia	
	mice [75];		Acceleration/reversal of carcinogenesis in at risk subjects/colon cancer patients	
	mice [78];		Coagulation modulation	
	healthy, sedentary elderly humans, RCT [79];		Progression of chronic kidney disease	
	middle-aged Japanese women, *n* = 74 [80];		Weight loss and improved insulin resistance	
	non-equol-producing menopausal Japanese women, *n* = 93 [81]		Significant carotid arterial stiffness reduction; improvement of cholesterol profile	
			Whole body bone mineral density increase in postmenopausal women	
			immune-modulation and anti-allergic properties	

## Data Availability

PubMed and Medline; the main national and international gastroenterological meetings online databases (e.g., United European Gastroenterology Week, Digestive Disease Week).

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
