# Peer review of "From Pre- and Probiotics to Post-Biotics: A Narrative Reviewâ€"

_ijerph, 2021, doi:10.3390/ijerph19010037_

Round 1
Reviewer 1 Report
The review article "From Pre- and Probiotics to Post-Biotics: A Narrative Review" is interesting and well-organized. It is a hot topic. It is not very very original but it is interesting because it is updated and well-structured. Moreover, it is easy to follow if you are outside the field.
Nevertheless, table 1 is not enough to understand the relevance of each study: conditions of the study, such as experimental model and number of subjects (N) should be added and each reference should be described separately.
Author Response
We thank the Reviewer for comments and suggestions.
We have replied point-by-point to his comments and suggestions.
The corrections made are in red within the text.
REVIEWER 1:
The review article "From Pre- and Probiotics to Post-Biotics: A Narrative Review" is interesting and well-organized. It is a hot topic. It is not very original but it is interesting because it is updated and well-structured. Moreover, it is easy to follow if you are outside the field.
Nevertheless, table 1 is not enough to understand the relevance of each study: conditions of the study, such as experimental model and number of subjects (N) should be added and each reference should be described separately.
We thank the Referee for these suggestions. We have added the required data to the Table.
Reviewer 2 Report
This manuscript is a review summarizing the literature data on the nomenclature and description of postbiotics, the potential mechanism of their action and application in the prevention/treatment of diseases, including the use of postbiotics in patients with COVID-19. The thematic scope of this manuscript corresponds to the subject of the Int. J. Environ. Res. Public Health journal. Therefore, in my opinion, this manuscript is eligible for publication in this journal. However, I believe that this manuscript should be slightly corrected and supplemented with missing data before it is released for publication:
- please update the LAB nomenclature according to the guidelines provided on the website: http://lactobacillus.ualberta.ca/. I believe that the term "Lactobacillus" can be changed to "lactobacilli " where possible.
- Line 182: are only anti-oxidant-enzymes important? there are also publications on BSH (bile salts hydrolase) activity in LAB.
- Line 224: Please note that the ratio of SCFAs is important for cholesterol homeostasis.
- Lines 263-266: is only vitamin B12 essential? What about the other B vitamins, and what about vitamin K, synthesized by the intestinal microflora and LAB? Please add some comment to the text.
- Lines 267-270: please pay attention to the bioactive peptides released as a result of the peptidolytic activity of bacteria.
- No studies on effects on the immune system? alleviation of allergy symptoms?
- Line 360: please correct the reference.
- Please verify reference number: 26, 38, 49, 52, 59, 68, 73, 74, 76, 78, 79, and 80.
Author Response
REVIEWER 2:
This manuscript is a review summarizing the literature data on the nomenclature and description of postbiotics, the potential mechanism of their action and application in the prevention/treatment of diseases, including the use of postbiotics in patients with COVID-19. The thematic scope of this manuscript corresponds to the subject of the Int. J. Environ. Res. Public Health journal. Therefore, in my opinion, this manuscript is eligible for publication in this journal. However, I believe that this manuscript should be slightly corrected and supplemented with missing data before it is released for publication:
please update the LAB nomenclature according to the guidelines provided on the website: http://lactobacillus.ualberta.ca/. I believe that the term "Lactobacillus" can be changed to "lactobacilli " where possible.
We thank the Referee for this suggestion. We have updated the nomenclature according to the suggested website.
Line 182: are only anti-oxidant-enzymes important? there are also publications on BSH (bile salts hydrolase) activity in LAB.
We thank the Referee for this suggestion. We have added the data and relative reference.
Line 224: Please note that the ratio of SCFAs is important for cholesterol homeostasis.
We thank the Referee for this suggestion. We have modified the sentence.
Lines 263-266: is only vitamin B12 essential? What about the other B vitamins, and what about vitamin K, synthesized by the intestinal microflora and LAB? Please add some comment to the text.
We thank the Referee for this observation. We have added the required data.
Lines 267-270: please pay attention to the bioactive peptides released as a result of the peptidolytic activity of bacteria.
No studies on effects on the immune system? alleviation of allergy symptoms?
We thank the Referee for this observation. We have added the required data.
Line 360: please correct the reference.
We thank the Referee for this suggestion. We have made the correction.
Please verify reference number: 26, 38, 49, 52, 59, 68, 73, 74, 76, 78, 79, and 80.
We thank the Referee for this suggestion. We have verified them.
Round 2
Reviewer 2 Report
I see that the Authors have improved the manuscript according to the suggestions of reviewers. I do not have more comments to this manuscript. In my opinion, it is suitable for publishing in an IJERPH journal in the present form.